# Postacute Laryngeal Injuries and Dysfunctions in COVID-19 Patients: A Scoping Review

**DOI:** 10.3390/jcm11143989

**Published:** 2022-07-09

**Authors:** Jérôme R. Lechien, Stéphane Hans

**Affiliations:** 1Department of Otolaryngology-Head & Neck Surgery, Foch Hospital, School of Medicine, UFR Simone Veil, Université Versailles Saint-Quentin-en-Yvelines (Paris Saclay University), 92150 Paris, France; prhans.foch@gmail.com; 2Department of Human Anatomy and Experimental Oncology, Faculty of Medicine, UMONS Research Institute for Health Sciences and Technology, University of Mons (UMons), 7000 Mons, Belgium; 3Department of Otolaryngology-Head & Neck Surgery, CHU Saint-Pierre (CHU de Bruxelles), 1000 Brussels, Belgium; 4Department of Otolaryngology, Elsan Polyclinic of Poitiers, 86000 Poitiers, France

**Keywords:** COVID-19, otolaryngology, larynx, laryngeal, laryngology, intubation, voice, head neck, surgery

## Abstract

Objective: To investigate post-acute laryngeal injuries and dysfunctions (PLID) in coronavirus disease 2019 (COVID-19) patients. Methods: Three independent investigators performed a systematic review of the literature studying PLID in patients with a history of COVID-19. The review was performed according to the Preferred Reporting Items for Systematic Reviews and Meta-Analyses Statement. Epidemiological, clinical, hospitalization features, laryngeal diseases, and voice outcomes were extracted from the included papers. Results: Eight papers met our inclusion criteria (393 patients) corresponding to five uncontrolled prospective and three retrospective studies. The most prevalent PLID were vocal fold dysmotility (65%), vocal fold edema (35%), muscle tension dysphonia (21%), and laryngopharyngeal reflux (24%). Posterior glottic stenosis (12%), granuloma (14%), and posterior glottic diastasis (12%) were the most common injuries. Most patients with PLID were obese and had a history of intensive care unit hospitalization, and orotracheal intubation. The delay between the discharge and the laryngology office consultation ranged from 51 to 122 days. The mean duration of intubation ranged from 10 to 34 days. Seventy-eight (49%) intubated patients were in the prone position. The proportion of patients requiring surgical treatment ranged from 39% to 70% (mean = 48%). There was an important heterogeneity between studies about inclusion, exclusion criteria, and outcomes. Conclusions: COVID-19 appeared to be associated with PLID, especially in patients with a history of intubation. However, future controlled studies are needed to evaluate if intubated COVID-19 patients reported more frequently PLID than patients who were intubated for other conditions.

## 1. Introduction

Coronavirus disease 2019 (COVID-19) was associated with more than 510 billion cases and 6,200,000 deaths as of 1 May 2022 [1]. According to studies, 5 to 20% of COVID-19 patients had severe-to-critical disease and required mechanical ventilation [2,3]. The proportion of survivors after severe or critical COVID-19 ranges from 20 to 62% regarding world regions [4,5,6]. The survivor follow-up highlighted that they may keep neurological, psychological, and systemic post-discharge sequelae [7]. Precisely, over the past few months, an increasing number of studies suggested that COVID-19 patients may report postacute laryngeal injuries and dysfunctions (PLID), especially after hospitalization in the intensive care unit (ICU) [8,9,10,11,12].

The aim of this scoping review was to investigate post-discharge PLID in COVID-19 patients.

## 2. Materials and Methods

The criteria for consideration of study inclusion were based on the population, intervention, comparison, outcome, timing, and setting (PICOTS) framework [13]. Data of the study were independently reviewed by three investigators (JRL, SH, and MC) who extracted findings according to the Preferred Reporting Items for Systematic Reviews and Meta-Analyses (PRISMA) checklist for systematic reviews [14].

### 2.1. Patient Population

Prospective and retrospective, controlled, uncontrolled, or randomized studies published between December 2019 and May 2022 were included if authors investigated PLID in COVID-19 patients. Studies only reporting symptom outcomes without laryngeal examination were excluded. Patients should have a positive diagnostic for COVID-19 by reverse transcriptase polymerase chain reaction (RT-PCR) testing or serology. The studies were published in English, Spanish, or French peer-reviewed journals. Case reports were not considered in the analysis. The type of study was classified according to the levels of evidence for prognostic studies (I–V) [15].

### 2.2. Intervention, Comparison, and Outcomes

The following outcomes were reviewed for each study: number of patients; mean age; gender; COVID-19 severity; intubation or tracheostomy history and characteristics; duration of hospitalization; post-discharge laryngeal/voice findings and methods of laryngeal outcome assessment. The Tool to Assess Risk of Bias in Cohort Studies developed by the Clarity Group and Evidence Partners was used by two authors (JRL & SH) for the bias/heterogeneity analyses of the included studies [16]. The bias analysis consisted of an evaluation of cofactors that may impact the conclusion of studies.

### 2.3. Timing and Setting

The patients had been cured of COVID-19 and were discharged from the hospital.

### 2.4. Search Strategy

The paper search was conducted with PubMed, Cochrane Library, and Scopus databases by three independent laryngologists (JRL, SH, and MC). Databases were screened for abstracts and titles referring to the inclusion criteria of the present study. Authors analyzed the full texts of the selected publications. The following keywords were considered for the literature search: ‘COVID-19’; ‘SARS-CoV-2’; ‘Larynx’; ‘Voice’; ‘Laryngeal’; ‘Complications’; ‘Intubation’; and ‘Otolaryngology’.

## 3. Results

### 3.1. Study Features

The present study did not require institutional review board approbation. Eight papers met our inclusion criteria, accounting for 393 patients (Figure 1) [8,9,10,11,12,17,18,19]. Five studies were uncontrolled prospective [8,10,12,17,19] and three publications were retrospective case-series [9,11,18]. Two studies were initially excluded because lack of laryngeal examination. The characteristics of studies are described in Table 1. The mean age of patients ranged from 42 to 63 years old. Patients consisted of individuals with a history of ICU hospital stay and orotracheal intubation [9,10,12,19]; non-intubated individuals [17]; or mixed populations [8,11,18]. Inclusion, exclusion criteria and comorbidities of patients are reported in Table 2. Comorbidity data were not reported in one study [17]. Studies included patients with the following comorbidities: hypertension (N = 124/237; 52%); tobacco consumption (N = 44/89; 49%); diabetes (N = 100/237; 42%); laryngopharyngeal or gastroesophageal reflux disease (N = 11/34; 33%); obesity (N = 55/207; 27%); asthma (N = 15/71; 21%); coronary disease (N = 15/88; 17%); chronic obstructive pulmonary disease (COPD) (N = 13/104; 13%); obstructive sleep apnea syndrome (N = 13/122; 11%) and stroke (N = 2/25; 8%) (Table 2). Exclusion criteria were specified in four papers, and included patients with laryngeal disorders or dysphonia prior COVID-19 and those without confirmation of the COVID-19 diagnosis (RT-PCR, serology) [11,12,17,18]. Azzam et al. also excluded patients with voice disorders occurred >1-month post-infection, or with a history of head and neck cancer or trauma [17].

Among patients presenting with voice disorders post-discharge, there were 184 females and 209 males. The mean body mass index was provided in two studies and was elevated (>25) [10,11]. The mean delay between the discharge and the laryngology office consultation ranged from 51 to 122 days [11,12,17,18,19].

The features of patients who were in ICU are presented in Table 1. Focusing on studies where authors reported findings of ICU patients, the mean duration of intubation ranged from 10 to 34 days, while the size of the tube ranged from 7 to 8. Seventy-eight (49%) intubated patients were in prone position during the ICU stay. There were 88/237 patients (37%) with a tracheostomy, which was removed after a mean duration ranging from 16 to 70 days.

### 3.2. Laryngeal Abnormalities

Irrespective of the definition and voice quality tools used, dysphonia was found in 70% of patients. Most patients had multiple chiefs of complaints (Table 1). PLID reported in studies are summarized in Table 3. Note that the prevalence of PLID was assessed in studies in which specific data of intubated patients were reported. Thus, the study of Neunheim et al. was excluded from this analysis according to the pooled information from non-intubated and intubated patients. Among COVID-19 patients with dysphonia in the post-discharge follow-up, the most prevalent PLID were vocal fold dysmotility, edema, laryngopharyngeal reflux, and muscle tension dysphonia (Table 3). The most prevalent laryngeal injuries included posterior glottic stenosis, granuloma, posterior glottic diastasis, and VF immobility. Sandblom et al. reported a positive association between the duration of ICU stay and the severity of swallowing disorders [10]. Among contributing factors, Hans et al. reported that prolonged intubation was associated with an increase in laryngeal injuries, e.g., posterior glottic stenosis [19], while Felix et al. observed that intubation tube size and prone position were both factors associated with laryngeal injuries [12].

Some authors reported the need for medical (botox or corticosteroid injections) or surgical (balloon dilatation or laser microsurgery procedures) treatments for the management of laryngeal injuries [11,19]. Overall, the proportion of patients requiring surgical approach ranged from 39% to 70% (mean = 48%) [11,19].

### 3.3. Bias Analysis

Bias analysis is reported in Appendix A. There was no study that compared the prevalence of PLID between COVID-19 discharged patients and those discharged from the hospital for another reason. Three studies were retrospective case-series (EL: IV) and five were prospective uncontrolled studies (EL: III). The patient populations substantially varied from one study to another according to the proportion of intubated versus not intubated patients, and therefore, the severity of the disease. Moreover, the delay between the discharge and the time of examination was not provided in three studies [8,9,10] and may vary from one study to another, which may lead to a comparison bias. Other factors may limit the comparison between studies, e.g., the variability in comorbidity prevalence, the proportion of tracheostomy, the duration of intubation or tracheostomy, and the methods used to evaluate the laryngeal function and PLID (Appendix A). Moreover, many important outcomes that may influence the development of PLID were not investigated in patients, including tobacco consumption, laryngopharyngeal reflux, or a history of previous cancer or radiation. No author reported medical post-discharge care (drugs), which may impact the development of some laryngeal injuries.

## 4. Discussion

Coronavirus disease 2019 was found to be associated with many otolaryngological disorders, including smell and taste dysfunctions [20], vestibular neuritis [21], parotitis [22], and facial paralysis [23] or paradoxical vocal fold movement [24]. The association between COVID-19 and laryngeal disorders was initially observed in a first-wave epidemiological study in which 26% of patients with mild-to-moderate COVID-19 reported dysphonia throughout the clinical course of the disease [25]. Since then, the follow-up of patients who were intubated or who had a tracheostomy in ICU suggested the occurrence of mid-term PLID [8,9,10,11,12]. The present review summarized the PLID found in COVID-19 patients. Many factors may limit the draw of a reliable conclusion.

First, most authors suggested that the prevalence of post-intubation or post-tracheostomy PLID was substantially high in COVID-19 patients who were discharged from the ICU. Our data suggested that the prevalence of bilateral or unilateral vocal fold motion disorders, vocal fold edema, or posterior glottic stenosis may reach 16% to 65% of cases. A recent systematic review reported that 13% to 31% of non-COVID-19 patients who were intubated in ICU had moderate-to-severe laryngotracheal injuries [26]. Precisely, grade 3 injuries, including stenosis, hypo/immobility of vocal folds, and/or arytenoid complex, were found in 13% of cases, whereas grade 2 injuries (e.g., hematoma, ulceration, edema, or granulation) concerned 31% of cases [26]. Authors reported that the prevalence of grade 3 and 2 injuries increased to 33% and 18% in patients with an average intubation duration >5 days, respectively [26]. Comparing these data with the findings of the present review, it could appear that COVID-19 patients may present higher rates of PLID but, according to the lack of controlled study, the draw of reliable conclusion remains difficult.

Second, many other biases may limit the interpretation of the study results. The primary bias is an inclusion bias because patients included in the studies were all recruited from otolaryngology or laryngology offices and, therefore, consisted of a dysphonic population. There was no epidemiological study that systematically evaluated the occurrence of voice disorders and PLID in all patients who were discharged from the hospital/ICU.

Third, the authors investigated some laryngeal disorders without consideration of other contributing factors that may be associated with the development of PLID. This is particularly the case for muscle tension dysphonia, which was reported as a prevalent condition in COVID-19 patients in two studies [8,18]. Muscle tension dysphonia may develop from gastric or environmental irritants, laryngitis, or even stress, among other conditions [27], which were not investigated in both studies [8,18]. A similar observation may be made for granuloma. Although intubation is an important cause of granuloma, other prevalent etiologies may play a key role in the development of granuloma such as reflux [28]. According to studies, the populations of studies considered in the present review reported high but different rates of comorbidities, which may be an additional limiting factor to precisely study the prevalence of PLID and its association with COVID-19. Indeed, some comorbidities may be associated more frequently with some PLID, such as reflux and posterior laryngeal edema; or diabetes and laryngotracheal stenosis [28,29]. Another factor that may impact the results of studies is the definition of PLID. We observed that the definition of some PLID may vary from one study to another. Precisely, the authors did not define similarly vocal fold hypomobility, which was the most prevalent PLID. The observation of hypomobility of vocal folds is still subjective and many authors did not provide information about the etiology (laryngeal nerve impairment versus arytenoid join ankylosis) [10,11,18]. These biases have to be considered in future studies that aim to investigate PLID in COVID-19.

The present scoping review globally included eight studies (393 patients), which considerably limits the drawing of reliable conclusions. The low number and the low evidence level of studies are, therefore, the primary limitation of this review. The lack of controlled study comparing both prevalence and features of post-intubation/post-tracheostomy PLID between discharged COVID-19 and individuals with a history of intubation or tracheostomy for another condition is another important limitation.

## 5. Conclusions

COVID-19 appeared to be associated with PLID in patients with a history of intubation or tracheostomy. However, it remains difficult to determine if the development of post-intubation or post-tracheostomy PLID is more frequent in COVID-19 patients compared with those with a history of intubation or tracheostomy for another condition. Future controlled studies are needed to compare the prevalence of post-intubation or post-tracheostomy PLID in both populations.

## Figures and Tables

**Figure 1 jcm-11-03989-f001:**
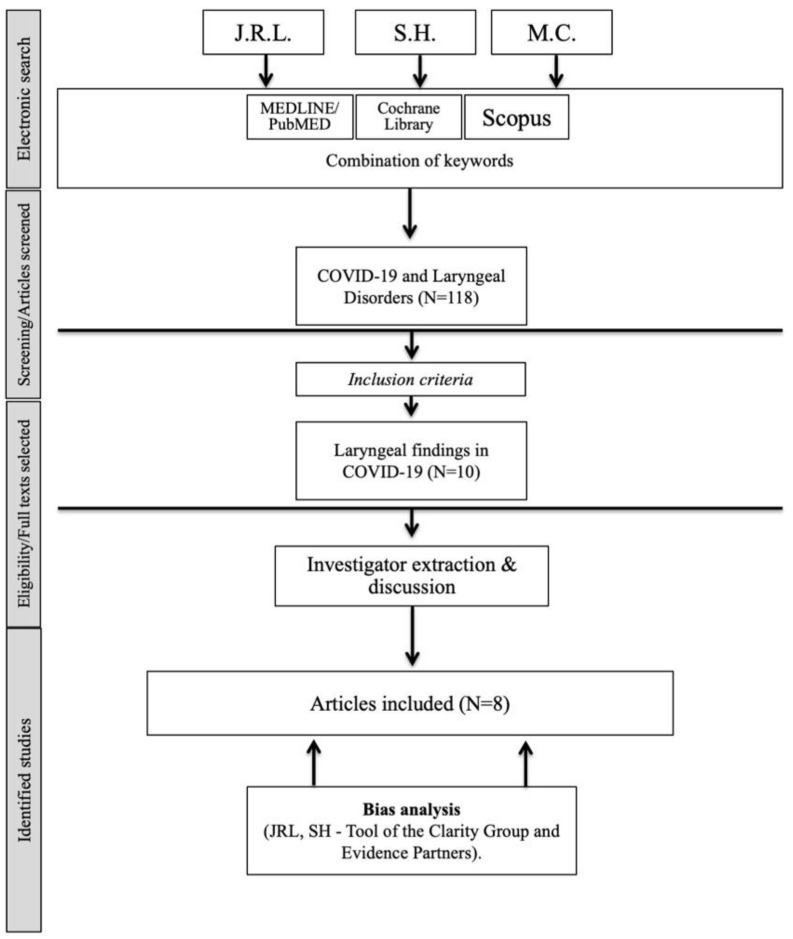
Flow chart.

**Table 1 jcm-11-03989-t001:** Studies investigating laryngeal disorders and injuries in COVID-19 patients.

Authors	Study	EL	Population	ICU Outcomes	Voice/Laryngeal Outcomes	Results	Conclusion
Naunheim [8]	Prospective	III	Gr1 = 13 dysphonic ICU	Intubation (13): 22 d	Disorder prevalence	Gr1-2 N, %	1.Most patients with dysphonia and history of COVID-19 had history of intubation. The occurrence of laryngeal lesions was found in post-intubated patients.
USA	Uncontrolled		Gr2 = 7 non-ICU patients	Tube size: 7.5	Voice disorders	3 (43)–9 (69)	
				Tracheostomy: 9	Breathing disorders	2 (28)–5 (38)	
			Age = 59 yo	Duration: 16 d	Stroboscopy abnormalities	17 (85)	
			F/M = 5/15	Proning: 9	Vocal fold immobility	8 (40)	
			BMI = NP		Posterior glottic stenosis	3 (15)	
			Delay = NP		Subglottis stenosis	2 (10)	
					Posterior glottic diastasis	2 (10)	2. Nine patients required procedural interventions; 4 in operating room.
					Laryngopharyngeal reflux	2 (10)	
					MTD	1 (5)	
					Intervention need	9 (45)	
Scholfield [9]	Retrospective	IV	N = 3 post-intubated ICU	Intubation (3): 34 d	Prevalence of		1. Subglottis stenosis may occur early in COVID-19 patients who were long-time intubated or tracheotomized.
UK	Case-series		Age = 49 yo	Tube size: 8	Subglottis stenosis	N = 3	
			F/M = 1/2	Tracheostomy: 3			
			BMI = NP	Duration: 30 d			
			Delay = NP	Tube: 7–9			
				Proning: N.P.			
Sandblom [10]	Prospective	III	N = 25 post-intubated ICU	Intubation (25): 10 d	FEES penetration	23 (96)	1. Vocal fold movement disorders are prevalent in post-intubated patients.
Germany	Uncontrolled		Age = 63 yo	Tube size: NP	Vocal fold dysmotility	19 (76)	
			F/M = 2/23	Tracheostomy: 20	Vocal fold immobility	2 (8)	
			BMI = 28	Duration: 30 d	Granuloma	2 (8)	2. There was a positive correlation between ICU hospitalization duration and dysphagia severity.
			Delay = NP	Tube: NP	Vocal fold hematoma	1 (4)	
				Proning: 12	Vocal fold ulceration	1 (4)	
Neevel [11]	Retrospective	IV	N = 18 dysphonic ICU	Intubation (18): 14 d	V-RQOL score (N = 14)	73	1. Most patients had multiple chief voice complaints.
USA	Case-series		N = 2 non-ICU patients	Tube size: 8	Intubated patients (N = 18):		
			Age = 50 yo	Tracheostomy: 10	VF motion impairments	9 (50)	2. Intubated patients reported high prevalence of laryngeal injuries.
			F/M = 12/12	Duration: 18 d	VF edema/erosion	7 (39)	
			BMI = 29	Tube: NP	Subglottis stenosis	4 (22)	3. Non-intubated patients reported tension muscle dysphonia (4), glottic edema (1), laryngitis (1), and unilateral VF paresis (1)
			Delay= 107 d	Proning: 10	Posterior glottic diastasis	4 (22)	
					Posterior glottic stenosis	3 (17)	
					Unilateral VF immobility	4 (22)	
					Unilateral VF hypomobility	2 (11)	4. Surgical/medical treatments were made in 10 and 4 patients.
					Bilateral VF hypomobility	3 (17)	
Felix [12]	Prospective	III	N = 95 post-intubated ICU	Intubation (95): 12 d	Laryngeal injuries	38 (40)	1. Laryngeal injuries were found in 40% of intubated patients.
	Uncontrolled		Age = 59 yo	Tube size: 7–8	Hyperemia	6 (6)	
			F/M = 44/51	Tracheostomy: 20	Granuloma	15 (16)	2. Tube size and prone position were contributing factors of laryngeal injuries.
			BMI = NP	Duration: NP	Posterior glottic stenosis	16 (17)	
			Delay = 100 d	Tube: NP	Unilateral VF immobility	1 (1)	
				Proning: 47			
Azzam [17]	Prospective	III	N = 106 non-intubated	-	Dysphonia	84 (79)	1. Dysphonia I in 79% of mild-to-moderate COVID-19 patients.
Egypt	Uncontrolled		Age = 42 yo		VF edema	42 (40)	
			F/M = 78/28		VF swelling	18 (17)	
			BMI = NP		Unilateral VF immobility	14 (13)	2. Various laryngeal findings were found in videostroboscopy.
			Delay = <30 d		Ventricular band edema	20 (19)	
Allisan [18]	Retrospective	IV	Gr1 = 31 intubated	Intubation (31): 17 d	Disorders	Gr1-2, *p*-value	1.COVID-19 may be associated with laryngeal injuries and disorders in intubated and not intubated patients.
USA	Case-series		Gr2 = 50 not intubated	Tube size: 8	Dysphonia	20, 38; NS	
			Age = 54 yo	Tracheostomy: 18	MTD	1-19; S	
			F/M = 32/49	Duration: 70 d	LPR	1-18; S	
			BMI = NP	Tube: 7–9	VF paresis	3-3, NS	2.Granuloma, posterior glottis stenosis, VF paresis, and tracheal stenosis were the most prevalent diseases.
			Delay = 122 d	Proning: NP	VF paralysis	5-3, NS	
					VF atrophy	3-6, NS	
					VF polyp	0-8, NS	
					Granuloma	8-0, S	
					Glottis insufficiency	4-3, NS	
					Arytenoid ankylosis	1-5, NS	
					Posterior/subglottis stenosis	5-0, NS	
					Tracheal stenosis	5-0, NS	
Hans [19]	Prospective	III	N = 43 intubated	Intubation (43): 10 d	Posterior glottic stenosis	14 (33)	1. Posterior glottis stenosis, laryngeal edema and granuloma were the most prevalent laryngeal findings.
France	Uncontrolled		Age = 52 yo	Tube size: NP	Laryngeal edema	10 (23)	
			F/M = 10/33	Tracheostomy: 8	Granuloma	8 (19)	
			BMI = NP	Duration: NP	Laryngeal necrosis	2 (5)	
			Delay = 51 d	Tube: 7–9	Posterior glottic diastasis	2 (5)	2. Prolonged intubation was associated with an increase of laryngeal injuries (posterior stenosis).
				Proning: NP	VF atrophy	2 (5)	
					Subglottis stenosis	1 (2)	

Table 1 reports the clinical study features, i.e., population characteristics, ICU outcomes and laryngeal abnormalities according to the patient types (ICU versus non-ICU). Abbreviations: BMI = body mass index; FEES = fiberoptic endoscopic evaluation of swallowing; EL = evidence level; ICU = intensive care unit; LPR = laryngopharyngeal reflux; M/F = male/female; MTD = Muscle tension dysphonia; NP = not provided; VF = vocal fold; V-RQOL = voice related quality of life; yo = years old.

**Table 2 jcm-11-03989-t002:** Inclusion, exclusion criteria, and comorbidities of patients.

Authors	Inclusion	Exclusion	Comorbidities
Naunheim [8]	Voice-related disorder	NP	Hypertension (11), Tobacco (9),
	patients		Diabetes (8), Asthma (4), Obesity (3),
			OSAS (2), COPD (1)
Scholfield [9]	-	-	Diabetes (2), Obesity (2),
			Hypertension (2), OSAS (1), LPR (1)
Sandblom [10]	Post-intubated patients	NP	Hypertension (16), Diabetes (11),
	with dysphonia		Obesity (8), OSAS (4), Stroke (2),
			Coronary disease (6)
Neevel [11]	Dysphonic patients	Dysphonia before COVID-19	Diabetes (9), Hypertension (9),
		No confirmation of COVID-19	Tobacco history (8), Asthma/COPB (5)
			Coronary disease (2)
Felix [12]	Post-intubated patients	Dysphonia before COVID-19	Hypertension (52)
	with dysphonia	No confirmation of COVID-19	Diabetes (41)
			Obesity (28)
Azzam [17]	Mild-to-moderate	Dysphonia before COVID-19	NP
	COVID-19 cases	No confirmation of COVID-19	
		>1-month delay post-COVID-19	
		Severe COVID-19	
		Laryngeal lesion before COVID-19	
		Chemo/radiotherapy, Head Neck	
		Trauma or cancer histories	
Allisan [18]	Dysphonic patients	Laryngeal disorders before	Reflux (10), tobacco (9), Diabetes (9),
		COVID-19	Asthma (6), Anxiety (6), Obesity (4),
			Hypertension (4), OSAS (2), COPD (3),
			Depression (2)
			Depression (2), Panic disorder (2)
Hans [19]	Post-intubated patients	NP	Hypertension (30), Diabetes (20),
	with dysphonia		Tobacco (18), Dyslipidemia (12),
			Obesity (10), Coronary disease (7),
			COPD (4), OSAS (4)

Abbreviations: COVID-19 = coronavirus disease 2019; COPD = chronic obstructive pulmonary disease; LPR = laryngopharyngeal reflux; NP = not provided; OSAS = obstructive sleep apnea syndrome.

**Table 3 jcm-11-03989-t003:** Prevalence of laryngeal disorders and injuries.

Laryngeal Disorders	Number/Total	Prevalence	References
VF dysmotility	28/43	65.1	[10,11]
VF edema	59/167	35.3	[11,17,19]
MTD	20/81	24.7	[18]
Laryngopharyngeal reflux	19/81	23.5	[18]
Ventricular band edema	20/106	18.9	[17]
Bilateral VF hypo/immobility	3/18	16.7	[11]
Posterior glottic stenosis	39/237	16.5	[11,12,18,19]
Granuloma	33/244	13.5	[10,12,18,19]
Posterior glottic diastasis or atrophy	17/142	12.0	[11,18,19]
VF polyp	8/81	9.9	[18]
VF immobility	29/325	8.9	[10,11,12,17,18]
Subglottis stenosis	13/145	8.9	[9,11,18,19]
Glottis insuffisiency	7/81	8.6	[18]
VF hypomobility	8/99	8.1	[11,18]
VF ulceration or necrosis	3/68	4.4	[19]

Abbreviations: MTD = muscle tension dysphonia; VF = vocal fold.

## Data Availability

Not applicable.

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
