# Peer review of "Postacute Laryngeal Injuries and Dysfunctions in COVID-19 Patients: A Scoping Review"

_jcm, 2022, doi:10.3390/jcm11143989_

Round 1

Reviewer 1 Report

Dear Author,

I read with interest the presented Review " Postacute Laryngeal Injuries and Dysfunction in COVID-19 patients".

Unfortunately there are some big problems in your Review. You describe the kind of laryngeal morbidity of 393 patients with a history of  COVID -19 infection from 8 manuscripts. Some of these patients  needed ICU care others not. It is not possible to know if this is a big number compared to patients with a history of COVID-19 WITHOUT any laryngeal morbidity, as we do not know any rate or overall number. Also it is not possible to state if there is a relationship to COVID or not.

Therefor I do not see the reason for this Review and can not see any possible conclusion.

Author Response

According to the comment of the reviewer, we changed Table 1, specifying the prevalence of disorder regarding the intubation or not. For the description of results and lesions related to intubation, we only focused on intubation patients. Through these modifications, we have only data about intubated patients, which responds to the reviewer comment.

Please see the modified Table 1.

The following studies reported data about ONLY intubated patients or details about intubated vs non-intubated patients, allowing the ability to perform specific intubation-related analyses:

-Scholfield et al.; -Sandblom et al.; -Neevel et al.; -Felix et al.; -Allisan et al.; -Hans et al.

We changed the result section considering the comment of the reviewer:

Results, p.2, last lines: “Focusing on studies where authors reported findings of ICU patients, the mean duration of intubation ranged from 10 to 34 days, while the size of the tube ranged from 7 to 8. Seventy-eight (49%) intubated patients were in prone position during the ICU stay. There were 88/237 patients (37%) with a tracheostomy, which was removed after a mean duration ranging from 16 to 70 days.

Laryngeal abnormalities

Irrespective to the definition and voice quality tools used, dysphonia was found in 70% of patients. Most patients had multiple chiefs of complaints (Table 1). PLID reported in studies are summarized in Table 3. Note that the prevalence of PLID was assessed in studies in which specific data of intubated patients were reported. Thus, the study of Neunheim et al. was excluded from this analysis according to the pooled information from non-intubated and intubated patients.”

Consecutively, for the Table 3, we therefore excluded the data of Nauheim et al. The findings of this study was only considered for non-intubation descriptive data (dysphonia prevalence for example).

Reviewer 2 Report

1. Tables should be arranged in order.

2. Some words are not in English.

3. The order of the titles should be noted.

4. The first abbreviation must be clear.

5. Phrase repeated in line144 to146.

6. Informed consent statement should be written appropriately.

7. References need to be adjusted.

8.The numbers of patients included were 393, but the  prevalence of PLID was not calculated by this number, how to explain it?  Besides, the denominators of the prevalence rates of these diseases were  different, how to compared the differences? 

Author Response

  1. Tables should be arranged in order.

We arranged tables as requested.

  1. Some words are not in English.

The spelling was proofread by a native speaker EN for the revised paper.

Precisely, we changed “hospitalization” in “hospital stay”.

  1. The order of the titles should be noted.

Done.

  1. The first abbreviation must be clear.

We checked all abbreviation and corrected.

-We specified the PRISMA abbreviation:

Methods, line 2: “Data of study were independently reviewed by three investigators (JRL, SH and MC) who extracted findings according to the Preferred Reporting Items for Systematic Reviews and Meta-Analyses (PRISMA) checklist for systematic reviews [14].”

-We specified yo: results, line 4: “The characteristics of studies are described in Table 1. The mean age of patients ranged from 42 to 63 years old.”

-We specified in results, line 12: “COPD: chronic obstructive pulmonary disease (COPD)”

  1. Phrase repeated in line144 to146.

We do not have the line number in the author version paper. We checked the manuscript to delete potential repetition.

  1. Informed consent statement should be written appropriately.

We specified, first line of the results: “The present study did not require institutional review board approbation.”

  1. References need to be adjusted.

We checked the reference number and style.

8.The numbers of patients included were 393, but the  prevalence of PLID was not calculated by this number, how to explain it?  Besides, the denominators of the prevalence rates of these diseases were  different, how to compared the differences? 

That was related to different analysis: we analyzed the prevalence of voice disorder in all COVID-19 patients in a first time. In a second time, we only considered the prevalence of laryngeal lesions in post-intubated patients, excluding studies focusing on non-intubated patients (Azzam et al.) or without specific description of data according to the intubation history (Neuheim et al.). Excluding these studies, the second analysis of laryngeal disorder prevalence was performed on lower patient number.

We changed the result section considering the comment of the reviewer:

Results, p.2, last lines: “Focusing on studies where authors reported findings of ICU patients, the mean duration of intubation ranged from 10 to 34 days, while the size of the tube ranged from 7 to 8. Seventy-eight (49%) intubated patients were in prone position during the ICU stay. There were 88/237 patients (37%) with a tracheostomy, which was removed after a mean duration ranging from 16 to 70 days.

Laryngeal abnormalities

Irrespective to the definition and voice quality tools used, dysphonia was found in 70% of patients. Most patients had multiple chiefs of complaints (Table 1). PLID reported in studies are summarized in Table 3. Note that the prevalence of PLID was assessed in studies in which specific data of intubated patients were reported. Thus, the study of Neunheim et al. was excluded from this analysis according to the pooled information from non-intubated and intubated patients.”

Consecutively, for the Table 3, we therefore excluded the data of Nauheim et al. The findings of this study were only considered for non-intubation descriptive data (dysphonia prevalence for example).

Reviewer 3 Report

I suggest minor changes mainly related to the Tables. Table 1 should appear before Table 2. Table 1: I suggest using the same order for the items in the "ICU Outcomes" and "Voice/Larynx Outcomes" columns. All table abbreviations should be defined in the table footnote. There are some errors in the references. The footnotes in Appendix 1 are difficult to read, I suggest bolding the column headings and revising all definitions (yes, no, probably yes, probably) not included

Author Response

I suggest minor changes mainly related to the Tables. Table 1 should appear before Table 2. Table 1: I suggest using the same order for the items in the "ICU Outcomes" and "Voice/Larynx Outcomes" columns.

We corrected, especially for the reference 8.

All table abbreviations should be defined in the table footnote.

Done.

There are some errors in the references.

We corrected and added the “et al.” in reference with >7 authors.

The footnotes in Appendix 1 are difficult to read, I suggest bolding the column headings and revising all definitions (yes, no, probably yes, probably) not included

Bolding was done.
We rewrote the appendix 1 footnotes: “Appendix 1 footnotes: The criteria used and the definition of rating (yes, probably yes, probably no, and no) were explained below.

For confounding factors, authors had to exclude some conditions: Yes=exclusion of patients with pre-COVID-19 laryngeal disorders prior the COVID-19 and assessment of other conditions associated with PLID (i.e. reflux, tobacco exposition, trauma or radiation histories); probably yes=exclusion of patients with pre-COVID-19 laryngeal disorders or assessment of other conditions associated with PLID; Probably no=exclusion of only some confounding factors; No=no information provided about exclusion criteria.

Population analysis: Yes= different analyses performed according to the intubation/tracheostomy status of patients. No=no provided information.

Delay: Yes= information about the delay between the discharge of hospital and the occurrence of PLID in the different populations of patients were available (if applicable; intubated vs non-intubated; tracheotomized vs non-tracheotomized). Probably Yes= details for the entire cohort were available; Probably no= Only delay between COVID-19 diagnosis and PLID occurrence were available; No=no information provided.

Intubation/tracheostomy details: Yes= full information for concerned patients were provided (duration, tube size, proning); probably yes= >50% of information were provided; probably no= <50% of information were provided; No=no information were provided.

Postdischarge care: Yes= information about postdischarge medication/care (e.g. speech therapy) that may influence the development of PLID were provided. No=no information provided.

Round 2

Reviewer 2 Report

Informed Consent Statement section are still unmodified.